# Upper Respiratory Tract Infections in Sport and the Immune System Response. A Review

**DOI:** 10.3390/biology10050362

**Published:** 2021-04-23

**Authors:** Antonio Cicchella, Claudio Stefanelli, Marika Massaro

**Affiliations:** 1Department for Quality of Life Studies, University of Bologna, 40127 Bologna, Italy; claudio.stefanelli@unibo.it; 2Institute of Clinical Physiology, National Research Council (CNR), 73047 Lecce, Italy; marika.massaro@ifc.cnr.it

**Keywords:** upper respiratory tract infections in sport, cytokines, catecholamines, nutrient, endurance sports, training loads

## Abstract

**Simple Summary:**

This review aims at clarifying the relationships of heavy training with the upper respiratory tract infections (URTI), a topic which has reach the public awareness with the recent outbreaks of Covid 19. The URTIs are quite common in several sport activities among athletes who undergo heavy training. Causes of URTI are still poorly understood, because can be related with innate and genetic susceptibility and with several environmental factors connected with training load and nutrition. The time course of the inflammation process affecting URTI after training, has been also reviewed. After a survey of the possible physiological and psychological causes (stressors), including a survey of the main markers of inflammation currently found in scientific literature (mainly catecholamines), we provided evidence of the ingestion of carbohydrates, C, D, and E vitamins, probiotics and even certain fat, in reducing URTI in athletes. Possible countermeasures to URTI can be a correct nutrition, sleep hygiene, a proper organization of training loads, and the use of technique to reduce stress in professional athletes. There is a lack of studies investigating social factors (isolation) albeit with Covid 19 this gap has been partially fill. The results can be useful also for non-athletes.

**Abstract:**

Immunity is the consequence of a complex interaction between organs and the environment. It is mediated the interaction of several genes, receptors, molecules, hormones, cytokines, antibodies, antigens, and inflammatory mediators which in turn relate and influence the psychological health. The immune system response of heavily trained athletes resembles an even more complex conditions being theorized to follow a J or S shape dynamics at times. High training loads modify the immune response elevating the biological markers of immunity and the body susceptibility to infections. Heavy training and/or training in a cold environment increase the athletes’ risk to develop Upper Respiratory Tract Infections (URTIs). Therefore, athletes, who are considered healthier than the normal population, are in fact more prone to infections of the respiratory tract, due to lowering of the immune system in the time frames subsequent heavy training sessions. In this revision we will review the behavioral intervention, including nutritional approaches, useful to minimize the “open window” effect on infection and how to cope with stressors and boost the immune system in athletes.

## 1. Introduction

The definition of “immune system response” is complex, because involving thousands of different factors, some orchestrating together and some acting independently or organized in local networks. These factors are mediated by several mechanisms working at the transcriptional, molecular, and systemic level. In general, the immune system is challenged by “immune exposure”. An immune exposure has been defined as: “the process by which components of the immune system first encounter a potential trigger” [1] or an epitope, the antigen’s part which cause the immune response with antibodies, T and B cells proliferation.

Due to the high demands of the sport, and probably, to the strongest immune system, in athletes, the immune response is, in some extent, different from that one happening in non-heavily loaded organisms [2,3], making the elite athlete more vulnerable to infections. A genetic predisposition to URTI-like symptoms in sportsmen athletes has been advocated to explain this phenomenon [2,3].

Further, different sports feature for deep differences in pathogen exposures (e.g., water sports vs. land sports), and in the environment where the sports take place (e.g., winter and mountain sport vs. hot environments sports). Additionally, the organization and the contents of training in different sports, determine different responses of the athlete immune system, as well as sex, age, genetics factors and level of qualification (e.g., amateur vs. elite athletes). The interaction between these factors be manifold, complex and mostly unknown. For example, a young athlete who competes in a sport where physical appearance is important (e.g., rhythmic gymnastics) undergone severe diet restrictions, associate with heavy training regimes, while weightlifters have a high caloric intake, associate with intense loads of relatively short duration, and so on. Additionally, swimming, due to high chlorinated and somewhat cold exposure, has been shown to be associated with upper respiratory tract infections [2,4]. The literature about the immune system in sport and respiratory tract infections and training are somewhat controversial [5,6,7]. 

Existing reviews on the topic of immune resistance/suppression in athletes have mostly focused on single specific aspects, on physiological and, at lesser extent, on the psychological of this relationship. The scope of this review is to critically compare the different theories about immune response in heavily trained athletes and the possible countermeasures including dietary and nutraceutical approaches, to boost the immune system to minimize the risk of respiratory tract infection, highlighting the emerging relationships between physiological, metabolic, biochemical and genetic aspect of immune suppression in athletes and its relationships with behavioral and psychological aspects. Psychological stress has been shown to influence the immune system increasing the susceptibility to respiratory infections. Too much life stress has been proven to be a major factor in decreasing body defenses [7,8]. Subjects who have undergone to major life stressors, have been shown to be more prone to cold [8,9].

The recent spread of COVID 19, has further evidenced the problem of immune depression due to psychological aspects linked to stress and social isolation [10,11]. Changes in lifestyles (self-isolation, and quarantine) affected the mental health of the athletes worsening nutrition habits, sleep, and healthy lifestyles in general [10,11]. Interestingly, the higher the athletic identity, the worse the reactions to the isolation. Athletes with a higher athletic identity showed to ruminate and catastrophize more [10,11] showing a lower cognitive emotion regulation. However, COVID 19 is a lower respiratory tract infection, and this review focuses on upper respiratory tract infections.

## 2. Methods

We performed an online literature search, using the database PubMed from inception of the database to July 2020 using the following keywords in different combinations: “upper respiratory tract infections and sport”, “training and immune system” “endurance and immunity” “sleep and immunity” “exercise and inflammation”, “upper respiratory tract infections”, “stress and training”, “training overload,” “nutrition and recovery” “performance,” “recovery,” “fatigue,” stress”. Titles and abstracts of the papers were read, and on the basis of the selection criteria, relevant articles were retrieved for review. In addition, the reference lists from both original and review articles retrieved were also reviewed. Inclusion criteria were: papers dealt with respiratory (upper and lower) tract in elite athletes, were studies performed with humans (with the exception of one relevant study), were in English language, and were both experimental and theoretical studies. A total of 137 relevant papers were found from which 91 dealing with URTI, immune system and nutrition, were selected for review. A flow chart reporting the search strategy is reported in Figure 1.

## 3. Results

### 3.1. Sport and Upper Respiratory Tract Infections

Existing literature about heavy exercise and the respiratory tract show conflicting results about heavy exercise and its association with suppressed mucosal and cellular immunity and increasing symptoms of respiratory tract infections (URTIs-like symptoms). During competition in the cold, the incidence of upper respiratory tract infection is obviously very high: 20 out of 44 (45%) athletes and 22 out of 68 (32%) staff members of Finnish team experienced symptoms of the common cold during a median stay of 21 days at Winter Olympic Games [12], showing athletes as more prone to illness by a 13%.This result is of course influenced by the environmental conditions, but also in absence of cold weather has been shown that athletes participating in marathon in normal or hot environments [13], showed a 2- to 6-fold increased URTI risk during the 1–2-week post-race. In a large group of 2311 endurance runners, were found nearly 13.0% who reported illness in the week after the Los Angeles Marathon race compared with 2.2% of control runners [13]. This was confirmed by other epidemiological studies in triathletes and in marathon and ultramarathon race events and/or during and after very heavy training periods [13,14,15].

The decrease in exercise performance after a URTI can last 2–4 days, and runners who unwisely start an endurance race with systemic RTI symptoms are 2/3 times less likely to complete the race [5,16].

### 3.2. Susceptibility to URTI and the “Open Window” Theory

Nieman et al. observed that athletes who ran more than 96 km/week doubled their odds for sickness (URTI) compared with those running less than 32 km/week. Nieman also concluded that after acute bouts of prolonged and high-intensity exercise, many components of the immune system are suppressed for times lasting until several hours or even days [17]. This has led to the ‘open window’ theory, which was described as the 1- to 9-h after an endurance trial, when the host’s defenses are decreased and the risk of URTI increases [18]. While lasting this ‘open window’ period, athletes should be advised to be aware and to remain isolated from the possible sources of infection. The hypothesis of a J shape relationship between exercise dose and susceptibility to URTI has been proposed by Shepard, which identified in too low or too high exercise load a long-term depressing effect on the immune system, with the heavier load be a major factor to predispose to illness [19]. Another study [20] proposed instead a sinusoidal (S shape) relationship between infection odd ratio and training load, being the infections high with low loads, lower with moderate exercise (advocating a protective role for moderate exercise), high in heavily loaded but not elite athletes, and low again in elite athletes, having elite athletes an innate resistance to infections [5,20]. However, studies which investigated the pathogen responsible for URTI symptoms were able to identify it only in around half of the cases investigated [21], thus supporting the hypothesis that exercise-induced inflammation itself, at least in some cases, is the cause of URTI. The protective role of moderate exercise, was confirmed by other studies, suggesting the use of moderate exercise as a preventive measure to avoid URTI [22].

## 4. Biochemical Correlates of URTI

Elite athletes prone to recurrent URTI show an altered/adverse cytokine response to effort in comparison with healthy athletes [23]. Cox et al. [23], comparing illness prone runners with no illness prone ones, showed that in illness prone subjects IL levels (IL-8, IL-10, and IL-1ra) were lower at rest (19–38%), while postexercise IL-10 and IL-1ra were 10–20% lower, and IL-6 was elevated by 84–185%. Thus at least a part of URTI susceptibility can be attributed to constitutional/genetic factors. It has been hypothesized that the protective effect of regular exercise against respiratory infections can partly be sequel to shifts in serum levels of innate immunity proteins levels, as observed during periods of intensive exercise [6,24].

Aerobic exercise performed at high intensities for long times, provoke a marked drop in plasma concentration of glucose and amino acids, which can result in immunodepression. The availability of nutrients before (storage), during and after strenuous exercise is essential for a right immune system control. This control is coordinated by nutrient sensors (i.e., AMPK and mTOR) and metabolic pathways (i.e., glycolytic or oxidative phosphorylation) in immune cells [25,26]. Thymus, the site of production of T cells, is one of the main organs under stress for immune response. Thymic activity is reduced by strenuous exercise [26,27]. Thymic production of T cells declines with age, and experimental results raised the concern that high intensity exercise into the 4th decade of life may have adverse consequences for athletes’ health [26,27]. Data suggest that the effects of physical conditioning could be mediated mainly by an effect on T cells [20,26,27]. Prolonged and exhausting exercise reduces type-1 T-cell number and their capacity to secrete interferon-γ, a pro-inflammatory cytokine, [27,28]. Additionally, thymic stromal lymphopoietin (TSLP), an epithelial cell-derived cytokine, exhibits both pro-inflammatory and pro-homeostatic properties, on the basis of the context and tissues in which it is expressed. It is demonstrated that TSLP can trigger the production of Th2 cytokines, such as IL-13 and IL-4 [28,29].

Moreover, intensive training is associate with a higher number of blood type-2 and regulatory T-cells, which are known to produce the anti-inflammatory cytokines, IL-4 and IL-10, respectively. This increases the risk of upper respiratory symptoms, potentially due to the cross-regulatory effect of interleukin-4 on interferon-γ production and immunosuppressive action of IL-10 [23,28]. Time course of the inflammatory response is a key factor in determine time windows who make the organism prone to infections. It is well known that WBC (white blood cells) increases acutely after strenuous exercise. Correspondingly, we observed an acute (after 1 h cycling at exhaustion >70% of VO_2_max with 3 bursts of 10 min >80% VO_2_max), but not chronic (after 1 month cycling) of WBC count from 6.27 ± 2.34 × 10^3^/μL to 9.01 ± 3.63 × 10^3^/μL in trained cyclists [30]. A recommendation for practice is that the risk of URTI, is higher until after the 1 or 2 weeks following a marathon [18,31]. 

Heavy weights workout can also be a triggering factor of URTI. The immune cells of injured muscle secrete pro-inflammatory cytokines such as interleukins (mainly IL-1, IL-8, IL-6) and TNFα (tumor necrosis factor α), triggering a cascade of downstream inflammatory signaling pathways where NFκB (nuclear kappa-light-chain-enhancer of activated B cells transcription factor) represents one of the most significant signaling molecule activated upon injury in skeletal muscle [28,29]. 

### 4.1. Environmental Factors

Environmental factors have a significant role in promoting respiratory tract infections. Air pollution has been shown to be a factor facilitating lung inflammation. Results from literature suggests that acute PM 2.5 with different concentration can cause different degrees of adverse effects on lung, especially in high (>500 μg/m^3^) concentrations [32]. Some benefit has been observed from low intensity aerobic interval training in impeding the oxidative stress and inflammation caused by the exposure to pollution [32], helping in removing pollutants from the lungs.

### 4.2. Nutrition and URTI

In the general population, the adoption of “a well-balanced diet” significantly contributes to a healthy lifestyle and promote the state of well-being [33,34]. Combined with physical activity, a well-balanced diet can significantly contribute to reach and maintain a correct body weight, reduce the risk of chronic diseases and promote an overall good health also supporting an effective immune system and providing protection against infections, cancers and other diseases [4,25,35]. 

An “immune-supportive” role for nutrition has been hypothesized also for high level athletes [25,36]. Clinical and epidemiological data have shown that inadequate nutrition may contribute to impaired immunity and makes athletes more susceptible to infection [34,37]. Energy-restricted diets are often advised in sports [34,38] where lean bodies are more and more often advocated to improve performance, such as running and cycling [20,39] and could be accompanied by macro- and micronutrient deficiencies [4,40]. On the other side, excesses or unbalance in specific nutrients, such as carbohydrates at expense of protein, dehydration and excessive consumption of nutritional supplements in athletes, can lead to negative outcomes in the immune response and may be, at least partially, responsible for an increased infection risk [4,25,40]. 

Since effective immune cell functions require adequate amount of water, glucose, proteins electrolytes, and micronutrients [36,40] as a logical consequence, meet the nutritional cellular and system requirements should be strategic in the maintaining of an effective immune system. On this assumption, in order to minimize URTI and optimize post exercise recovery, nutritional intervention measures are often adopted by athletes and supporting teams as countermeasures to immunodepression. However, among the plethora of nutrients and nutritional strategies available, only a few of them have so far shown positive significant effects in maintaining athlete immune health. In the following sub-sections, we summarize the evidence gathered for some of the most promising approaches including, a nutritional intervention based on carbohydrates, amino acids, polyunsaturated fatty acids, some minerals, plant antioxidants, and vitamin D. 

### 4.3. Carbohydrates

Carbohydrates represents an “essential” fuel for most body cells including immune cells. On a nutritional point of view, the recommended daily intake of carbohydrates for athletes who train for more than 3 h a day is between 6 and 12 g/kg of body mass, with additional intakes of 30–60 g/hour during exercise lasting 1 h or more [41]. These recommendations aim to restore muscle and liver glycogen stores before exercise and maintain blood glucose levels upon exercise and ensure sufficient glucose availability for skeletal muscle contraction. However, carbohydrate availability seems also to exert a role in limiting the exercise-induced immune dysfunction. Glucose acts as a direct energizing substrate for immune cells [41,42]. Furthermore, since both catecholamines and cortisol have potent modulatory effects on immune function [41,42], increasing carbohydrate availability indirectly reduces the stress hormone response to the exercise, thereby limiting exercise-induced immune impairments [43,44,45].

While diets enriched in carbohydrate administered prior to exercise are associated with a dampen in the cytokine release, including IL-6 and IL-10 [44,45] they do not exert any beneficial effects on resting and post-exercise immune cell functions. Indeed, both high and low carbohydrate diets, administered for several days before exercise, have been associated with similar levels of bacterially stimulated neutrophil degranulation and mitogen-stimulated lymphocyte proliferation and resulted in a similar magnitude of impairment in the post-exercise [46,47]. The consumption of carbohydrate during the exercise attenuates the rise in plasma cytokines [47,48] and the trafficking of leucocytes, thus preventing the exercise-induced drop in neutrophil degranulation [46,48] and cause an increase in the burst of neutrophil respiratory activity [49]. Further, consuming carbohydrates while performing prolonged exercise, has the effect to prevent the decrease in anti-viral Type 1 helper T cells (number and percentage) and the suppression of IFN-γ production [46,48,50]. Carbohydrate consumption while exercising, diminishes the typical decreases in T lymphocyte proliferation following mitogen or antigen stimulation post-exercise [51]. However, in terms of URTI events, in a study performed on 98 runners consuming placebo or carbohydrates during a marathon there were not observed significant difference among the groups in the reported illness events in the fifteen days after the race [49]. In conclusion, carbohydrates represent a potential countermeasure against exercise-induced inflammation especially if consumed as a supplement during exercise. However, evidence that their ingestion may be effective at reducing URTI events need to be further and more systematically investigated. In Table 1 are summarized the findings of experimental studies related to inflammatory markers in heavily trained athletes. Papers reporting results in non-heavily trained subjects, has been omitted.

### 4.4. Fatty Acids

Fatty acids are long-chain hydrocarbons that can be classified into four categories: saturated fatty acids, monounsaturated fatty acids (MUFAs), polyunsaturated fatty acids (including Ω-3 and 6) polyunsaturated fatty acids—PUFA—and trans fats. More than 20 types of fatty acids are found in different foods, most of all vegetable oils, seeds, nuts and fish oils [52,53]. 

The principal roles of fatty acids are as energy sources and membrane constituents [53,54]. However, certain fatty acids have additional roles, such as precursors for the synthesis of bioactive lipid mediators, regulators of membrane and intracellular signaling processes including the activation of transcription factors and the modulation gene expressions [52,54]. Through these different actions, fatty acids are able to influence many key cellular functions including immune and inflammatory responses [55,56]. A huge mass of clinical and experimental evidence suggests that saturated and Ω-6 PUFA may promote inflammatory processes while Ω-3 PUFA eicosapentaenoic (EPA) and docosahexaenoic fatty acids (DHA) clearly exert anti-inflammatory activities [55,56]. EPA and DHA, independently have an anti-inflammatory effects [55], and has been shown a their counter-effects on classic inflammatory stimuli (endotoxin as well as saturated fatty acids and *n*-6 PUFAs) [56,57]. Both EPA and DHA became part of the cells’ membranes, partly replacing arachidonic acid. In this way, they result in decreased secretion of pro-inflammatory and immunosuppressive Ω-6-derived lipid mediators [54,58]. Must be noted that both EPA and DHA function as substrates involved in the synthesis of potent mediators able to reduce the inflammation and enhance the immune function: resolvins, protectins and maresins [58,59]. EPA and DHA also decrease the inflammatory responses caused by neutrophils, macrophages and endothelial cells by reducing the activation of the pro-inflammatory transcription factor NF-κB and activating the peroxisome proliferator activated receptor γ [58,60]. Finally, within antigen presenting cells, T-cells and B-cells, EPA and DHA specifically act by regulating key signaling events within the cell membrane [60]. Based on this data, the evaluation of the effects of saturated and Ω-6 PUFA on immune functions and inflammation in the context of physical exercise has been rather limited. Rather, a considerable number of studies have been performed aimed to evaluate the protective role of EPA and DHA, on immune function and inflammation in both sedentary and trained individuals. These studies have tested moderate (< 1.0 g/day) to high (4 g/day) doses of EPA plus DHA administered daily for one week to several months. Some studies have reported that supplementation with EPA and DHA decreases muscle soreness and inflammation in both male and female trained and untrained individuals [61,62]. In particular, in untrained individuals, EPA and DHA have been reported to diminish the exercise-induced elevation in key pro-inflammatory cytokines including TNF-α and IL-6 [60,61]. Interestingly, this effect was also seen with DHA alone [61,63]. Thus, the majority of studies suggest that Ω-3 PUFA are effective dampers of inflammatory response induced by exercise and this may translate to less damaged muscles. In contrast, the evidence supporting a role of Ω-3 PUFA in the prevention of URTI events remains scarce and inconsistent [60,61]. Although the frequent methodological limitations (in the study designs and small number of enrolled subjects) make it difficult to draw firm conclusions, EPA and DHA seems to decrease the exercise-induced inflammation, muscle damage and soreness in non-sportsmen. At present time, is not clear if EPA and DHA are able to influence the inflammation process or the response function in trained individuals. Larger studies of longer duration (several weeks to months) are necessary to explore better the effects of Ω-3 PUFAs, on inflammation and immune function in sportsmen. 

### 4.5. Vitamin D

Emerging research highlights for vitamin D a key role in both innate and acquired immunity [60]. Vitamin D acts as part of a heterodimer together to the vitamin D receptor (VDR) and the retinoid X receptor. Upon interaction, the newly constituted complex acts as a transcription factor by binding to the promoter regions of all those genes bearing specific vitamin D response elements [62,63]. Many cells of the immune system including monocytes, macrophage, neutrophils and T and B lymphocytes contain the VDR and also express the enzyme 1-α hydroxylase, which is responsible for hydroxylation of 25-(OH) D to its active form 1,25-(OH)2 D [62,64]. In particular, in the immune cells, vitamin D up-regulates gene expression of broad-spectrum anti-microbial peptides (AMP) that act as regulators in innate immunity [64] and immunomodulatory of T and B lymphocytes in acquired immunity [65,66]. In both trained and untrained individuals’ variations in vitamin D concentrations have been shown to affect the immune system effectiveness [67]. Studies in athletes [67,68,69] and the general population [67] have reported negative associations between vitamin D concentration and incidences of URTI. In outdoor male and female athletes, vitamin D concentrations have been negatively associated with the number of acute URTI events [67]. The cut-off point for contracting illness seems to be 95 nmol/L since athletes with circulating concentrations lower than this value had one or more episodes of illness whereas those with higher concentrations had one or fewer URTI episodes [69,70]. A similar study in endurance athletes reported that athletes showing circulating vitamin D concentration lower than 30 nmol/l experienced more URTI episodes than those reporting vitamin D concentrations over 120 nmol/l [70,71]. Finally, a recent intervention study performed in athletes engaged in regular sports training including rugby, volleyball, swimming, triathlon, cycling, and racquet, resulted that 14-week supplementation with 5000 IU per day of vitamin D3, during winter training, increased the SIgA levels [67,69]. The consensus for studies of elite athletes is that low levels of salivary IgA and/or secretion rates, low pre-season salivary IgA levels, declining levels over a training period, and failure to recover to pre-training resting levels, are associated with an increased risk of URTI [49]. In contrast, any association of URTI with vitamin D supplementation was found in elite rugby players and rowers [72]. Although these data are suggestive of a protective role by vitamin D in terms of URTI incidence, only placebo-controlled studies, more properly designed, will consent to definitively confirm the effectiveness of vitamin D prescription in promoting health and in preventing URTI in athletes.

### 4.6. Other Nutrients

Vitamin C and E have also been advocated, has as protective substances, thus fruit consumption [72,73,74]. Recently, intestine microbiota was indicated to have a potentiation role on the immune system [75,76]. Microbiota stimulates T cells and neutrophils, inducing a pathogen spreading control, and B cells. Nutrition countermeasures to exercise stress, probiotics (a derived of dairy products, mainly lactobacillus) has been recently investigated and recommended to improve gut microbiota [75,76]. However, the link between gut microbiota, mucosal immunity and exercise stimulation has not yet fully explored, leading to several possibilities for further research in the exercise immunology. Iron is ubiquitous in human body because linked with red cells. Vitamin C is essential for iron absorption. Iron and folate have been shown to strength the immune system in athletes, at least because endurance athletes are prone to iron deficiency. A sufficient iron supply is necessary by the host for trigger an effective immune response. Iron deficiency has been suggested to impact upon cell-mediated immunity. Iron was suggested to have an effect upon cell-mediated immunity and endurance and ultra-endurance athletes are an at-risk population for iron deficiency [34,37,77].

## 5. Strategies to Boost Immune System in Heavily Trained Athletes

Recovery procedures and training load distribution are the most obvious strategies to boost the body’s response system in athletes. Apart of the many nutritional suggestions, there are less explored and quite inexpensive systems to improve immunity in heavy training athletes. First one is the organization of the training. The tetradic system proposed in the ancient times [78] warning the athletes and the trainers about the organization of the schedule to avoid overloads. As stated by Philostratus: “The first day of introductory mild-intensity training, the second day is dedicated to strenuous exercise, followed by a day of low intensity, and another day of mild intensity exercise “. Proper sleep was also recommended together with regulating life habits and proper diet. Thus, the risks of overtraining in lowering the body defenses were yet well known in ancient time. A proper sleep is necessary to boost the body’s response to inflammation, but a question arises about which is a “proper” sleep. Sleep can be characterized both from duration than quality. For example, sessions placed in late evening hours, due to the activation of the adrenergic system (hypothalamus-pituitary-adrenocortical axis), has been proven to be detrimental to a good sleep and to a decrease in total sleep time and slow-wave sleep and REM sleep (the restoring sleep), while moderate exercise increase REM sleep [79]. The susceptibility window to infections after heavy training is a temporary phenomenon than can last from several hours to days [80], so, another important preventive measure to preserve the body in the susceptibility window, is to adopt physical measures during the recovery phases, for example hyperthermia (saunas) in the cold environments has been shown to be an effective method. Repetitive mild hyperthermia has been proven to be effective in elevating CD56(+), a factor of cells adhesion and growth, NKT (natural killer cells, which represent about 15% of total T cells) and B and T cells after 7 days of daily exposure at 40 degrees [81,82]. After exercise, cryotherapy has shown to have some effect on recovery [80], lowering peripheral inflammation. Much evidence exists about the efficacy of massage in improving the immune response. Recent studies in animals [83] and in men [84] shown the efficacy of massage in boosting effects on T cell repertoire and in decrease noradrenergic innervation of lymphoid organs. Lifestyle’s intervention and psychological methods has also been proven to be effective in boost the immune system, for example meditation and methods to copying with life stressors [85,86,87]. Quite simple activities, such as breathing control can be helpful in reducing stress [88,89]. An emerging measure to boost immune system, is the so-called nature-based therapy. Using relaxing environment to decrease stress, ameliorating subjective (stress scales) and objective (lower catecholamines) has proven to be an effective way to restore and improve the body’s defenses [90]. In this respect, social stress is known to be associated with a low immune response [91], and social situation of menace, panic or continuous pressure, can seriously impair the immune system. High level athletes must cope with high social pressure and thus stress, and this is also a possible concurrent cause of immunodepression. 

## 6. Conclusions

Immunity is a complex interaction between organs and the environment, mediated by several genes, receptors, molecules, hormones, cytokines, antibodies, antigens, inflammation substances which in turn relate to psychological factors. The immune system response of heavily trained athletes has been theorized to follow a J or S shape dynamics at times, high loads being effective in modifying the immune response elevating the biological markers of immunity and the organism susceptibility to infections. The cascade of inflammatory markers of inflammation has as the main player in the T cells systems. Training in a cold environment put the athletes at risk and every manageable and affordable countermeasure must be considered by coaches and athletes. Athletes, who are considered healthier than the normal population, are in fact prone to infections of the respiratory tract, due to lowering of the immune system in the time frames subsequent heavy sessions. Apart from behavioral intervention to minimize the “open window” effect on infection, some recommendation can be found in the scientific literature on how to cope with stressors and potentiate the immune system in athletes which are on heavy training or after the heavy competition. Key factors are the progression in training loads, a proper placement of sessions, and allowing a proper recovery. Relatively simple measures can be adopted, such as sleep hygiene and increase personal hygiene, reducing exposure to possible factors of infections, and proper nutrition rich in vitamins, hot beverages in winter, sauna, massage as proper recovery intervention can be helpful in preventing illness together with vaccination. While some correlation between psychological stressors and URTI are known (effect of major life’s stressors and social constraints), social factors are also emerging important determinants of susceptibility to inflammation (isolation). Environmental pollution and cleanliness are also emerging causes of URTI together with the availability of food with high content of high-quality nutrients and thus economic factors (income). A limitation of our review is we did not explore fully the genetics of URTI, especially in relationships to psychological factors. The study can be of benefit for coaches and athletes interesting in preventing URTI in their daily practice. Further research is needed in the modelling of the interaction between physiological and social factors involving the sphere of humanities sciences into the model of the immune response. 

## Figures and Tables

**Figure 1 biology-10-00362-f001:**
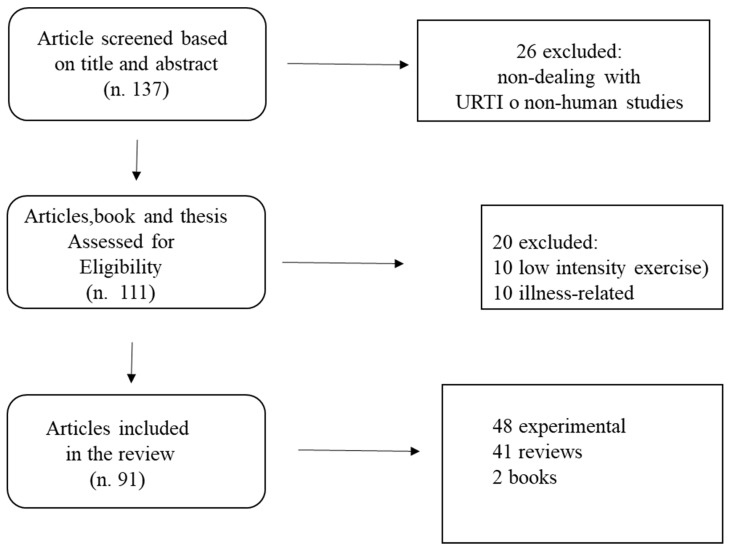
Flow chart of the search strategy (URTI: Upper Respiratory Tract Infection).

**Table 1 biology-10-00362-t001:** Biochemical/biological parameters measured in literature in heavy trained athletes (CHO: carbohydrates ingestion).

Author	Biochemical Parameter Measured	Population and or Treatment	Major Findings
Cox et al. 2007 [23]	IL-2, IL-4, IL-6, IL-8, IL-10, IL-12, and IL-1-RA	Illness prone vs. non illness prone athletes	IL-8, IL-10, IL-1RA lower at rest in illness-prone runners IL-10, 1-RA lower post exercise IL-6 strongly elevated post exercise
Kurowski et al. 2018 [24]	HSPA1,IL-1RA sCD14	Speed skaters	sCD14 and IL-1RA elevated in skaters vs. control. HSPA1 elevated in winter.
Prieto et al. 2014 [26]	TREC, CDT4, CDT3, CDT8, T-cells	Triathletes	Reduced compared to controls. CDT3 unvaried.
Bishop et al. 2001 [41]	Cortisol, IL-6, TNFα	Heavy cycling (CHO)	CHO ingestion reduced levels of cortisol and IL-6 but not TNFα
Nehlsen et al. 1997 [43]	IL-6, Ll-10, IL-1RA	Marathon (CHO)	CHO ingestion attenuates cytokine levels
Cannarella et al. 1997 [44]	IL-6, IL-1RA	Marathoners (CHO)	Lower IL-1RA after run. No increase in IL-1beta and IL-6.
Bishop et al. 2003 [45]	Neutrophil elastase content	Cyclists (CHO)	Blood levels Preserved by CHO ingestion
Scharhag et al. 2002 [46]	rhodamine(123), neutrophils	Cyclists (CH0)	Dimished after exercise with CHS
Lancaster et al. 2004 [47]	IFN-gamma positive CD4 + and CD8+ T lymphocytes	Cyclists (CHO)	CHO prevented the decrease of IFN-γ + CD4 and CD8+ T lymphocytes and the suppression of IFN- γ production
Nieman et al. 2002 [48]	Salivary IgA	Marathon (CHO)	The sIgA decreased in runners following a competitive marathon was not influenced by carbohydrate ingestion

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
