# Peer review of "Upper Respiratory Tract Infections in Sport and the Immune System Response. A Review"

_biology, 2021, doi:10.3390/biology10050362_

Round 1

Reviewer 1 Report

This is a very nice and beautiful written review showing the impact of sport on the immune system.

There are only very few grammatical errors throughout the manuscript, e.g. line 192-193 …. are have….

And phrases that should be rephrased such as Line 314.  Thus, yet in ancient time were well know the risk of overtraining.in lowering the body defenses,

Other than that, the only thing that I miss and which would further improve this review are to include literature on nutritional intervention with iron and folate that have a great influence on the immune system.

Author Response

Thank you very much for your time and effort spent to review this paper. We appreciate your suggestions, and modified the paper according to your instructions. Below, point by point what we have made. All the correction are evidenced in yellow in the manuscript text.

Reviewer  1.

here are only very few grammatical errors throughout the manuscript, e.g. line 192-193 …. are have….

We corrected some minor grammar errors all along the text. The text has now by revised by an English native expert.

And phrases that should be rephrased such as Line 314.  Thus, yet in ancient time were well know the risk of overtraining.in lowering the body defenses

Yes, we rephrased the sentence according to your suggestion.

Other than that, the only thing that I miss and which would further improve this review are to include literature on nutritional intervention with iron and folate that have a great influence on the immune system.

Iron is ubiquitous in human body, and of course has a direct influence also on URTI prevention. Iron and folate are yet referred  in some paper we reviewed it, reviewed a one more paper on the topic,  and we added this part to the text in the section “other nutrients”.

Reviewer 2 Report

The review provides current and interesting insight into the field of mutual dependencies between exercise and infection susceptibility, taking into consideration various additional co-factors. The authors interestingly describe the aspect of cytokine production under conditions altered by exercise. I would suggest some additions and amendments to this part of the manuscript, which are listed below.

  1. There is one important issue in the context of infection susceptibility after exercise. Studies designed to identify the pathogen responsible for URTI symptoms failed to achieve it in around half of the cases (e.g. Spence et al. Med Sci Sports Exerc. 2007 Apr;39(4):577-86. doi: 10.1249/mss.0b013e31802e851a.). This gives rise to the supposition that exercise may enhance inflammation and what is regarded as symptoms of URTI, is merely an exercise-induced inflammation without apparent pathogen to be held responsible. I suggest authors include this remark in their review.
  2. The authors cite paper by Cox et al [ref. # 15], but only mention briefly the altered cytokine response. These findings, however, are considered as one of the crucial ones indicating that inflammation and infections can be enhanced to different extent in part of the athletes, although they perform exercise at comparable levels as those who are not URTI-prone. I suggest some more detail into this cytokine dysbalance is included into the manuscript.
  3. In relation to previous point, anti-inflammatory proteins (e.g., CC16, IL-1ra) had been also studied in the context of exercise load and infection susceptibility suggesting that protective effect of regular exercise against respiratory infections can partly be sequel to shifts in serum levels of innate immunity proteins levels, as observed during periods of intensive exercise training. [Refs: Kurowski et al. Respiratory Research 2014, 15:45 &  Arch Med Sci. 2018 Jan;14(1):60-68. doi: 10.5114/aoms.2017.69438
  4. The authors begin their review with mentioning the COVID-19 conundrum in competitive exercisers. It would be suggested that the topic is referred to again in the text, considering that recommendations on maintaining immune health during lockdown periods have been issued, addressing also various nutritional and lifestyle aspects (e.g., Yousfi et al. Biol Sport. 2020;37(3):211–216

Minor issues:

Line 109: the reference seems not to be matched properly – should it be rather no. 15 instead of 14, please double-check!

The authors may consider putting some key messages (regarding e.g., cytokines, influence of nutritional factors or others) into tables. This would add to the clarity of the message.

Author Response

Thank you very much for your time and effort spent to review this paper. We appreciate your suggestions, and modified the paper according to your instructions. Below, point by point what we have made. All the correction are evidenced in yellow in the manuscript text.

There is one important issue in the context of infection susceptibility after exercise. Studies designed to identify the pathogen responsible for URTI symptoms failed to achieve it in around half of the cases (e.g. Spence et al. Med Sci Sports Exerc. 2007 Apr;39(4):577-86. doi: 10.1249/mss.0b013e31802e851a.). This gives rise to the supposition that exercise may enhance inflammation and what is regarded as symptoms of URTI, is merely an exercise-induced inflammation without apparent pathogen to be held responsible. I suggest authors include this remark in their review.

The paper has been updated with the reference of Spence et al., and in the text has been added this information,

The authors cite paper by Cox et al [ref. # 15], but only mention briefly the altered cytokine response. These findings, however, are considered as one of the crucial ones indicating that inflammation and infections can be enhanced to different extent in part of the athletes, although they perform exercise at comparable levels as those who are not URTI-prone. I suggest some more detail into this cytokine dysbalance is included into the manuscript.

This information has been added to the text, and the cytokine disbalance has been discussed.

In relation to previous point, anti-inflammatory proteins (e.g., CC16, IL-1ra) had been also studied in the context of exercise load and infection susceptibility suggesting that protective effect of regular exercise against respiratory infections can partly be sequel to shifts in serum levels of innate immunity proteins levels, as observed during periods of intensive exercise training. [Refs: Kurowski et al. Respiratory Research 2014, 15:45 &  Arch Med Sci. 2018 Jan;14(1):60-68. doi: 10.5114/aoms.2017.69438

Thank you very much for this important suggestion. We reviewed the paper of Kurowski, and added it to the paper, discussing the topic.

The authors begin their review with mentioning the COVID-19 conundrum in competitive exercisers. It would be suggested that the topic is referred to again in the text, considering that recommendations on maintaining immune health during lockdown periods have been issued, addressing also various nutritional and lifestyle aspects (e.g., Yousfi et al. Biol Sport. 2020;37(3):211–216.

We added some reference to the Covid topic and discussed it.

Minor issues:

Line 109: the reference seems not to be matched properly – should it be rather no. 15 instead of 14, please double-check!

We corrected some mismatch all along the paper text.

The authors may consider putting some key messages (regarding e.g., cytokines, influence of nutritional factors or others) into tables. This would add to the clarity of the message.

We added a summary table of experimental studies dealing with biochemical/biological parameters .

Round 2

Reviewer 2 Report

I am satisfied with Authors' amendments and explanations. My only suggestion, for the sake of the clarity of the message, is to replace (in line 142) 'URTI" with "URTI-like symptoms, since these are not infections (no pathogen detectable).